# A Systematic Review of Qualitative Research Literature and a Thematic Synthesis of Older LGBTQ People’s Experiences of Quality of Life, Minority Joy, Resilience, Minority Stress, Discrimination, and Stigmatization in Japan and Sweden

**DOI:** 10.3390/ijerph20136281

**Published:** 2023-07-03

**Authors:** Anna Sofia Bratt, Ann-Christine Petersson Hjelm, Matilda Wurm, Richard Huntley, Yoshihisa Hirakawa, Tsukasa Muraya

**Affiliations:** 1Department of Psychology, Faculty of Health and Life Sciences, Linnaeus University, 35195 Växjö, Sweden; rh222if@student.lnu.se; 2Department of Business Studies, Commercial Law, Uppsala University, 75120 Uppsala, Sweden; ann-christine.hjelm@fek.uu.se; 3School of Law, Psychology and Social Work, Örebro University, 70281 Örebro, Sweden; matilda.wurm@oru.se; 4Department of Public Health and Health Systems, Nagoya University, Nagoya 466-8550, Japan; y.hirakawa@med.nagoya-u.ac.jp; 5Faculty of Design, Kyushu University, Fukuoka 815-8540, Japan; muraya@design.kyushu-u.ac.jp

**Keywords:** aging, aged, sexual and gender minorities, quality of life, stereotyping, social discrimination

## Abstract

There is a lack of research on older lesbian, gay, bisexual, transgender, and queer/questioning (LGBTQ) adults. This systematic review aimed to synthesize Japanese and Swedish qualitative research on LGBTQ adults aged 60 years or older following the Preferred Reporting Items for Systematic Reviews and Meta-Analyses (PRISMA) guidelines. Japanese and Swedish articles, published in English, were searched across ASSIA, CINAHL, Medline, PsychINFO, PubMed, Scopus, and Sociological Abstracts databases. Additional searches were conducted to include studies in Japanese or Swedish. There were no papers from Japan, whereas five from Sweden were reviewed. One article was excluded due to the wrong phenomenon. Four articles were included, involving 48 participants aged 60–94 years. We summarized the findings using a deductive thematic synthesis. Two major themes emerged: (a) quality of life, minority joy, and resilience (positive aspects), and (b) discrimination, stigmatization, and minority stress (negative aspects). The participants wished to be acknowledged for their own assets and unique life histories, and to be treated as everyone else. They emphasized the importance of knowledge of LGBTQ issues among nursing staff, so that older LGBTQ people are treated in a competent and affirmative way. The study revealed several important topics for understanding older LGBTQ adults’ life circumstances and the severe lack of qualitative studies in Japan and Sweden.

## 1. Introduction

The proportion of people aged 60 years and older is increasing rapidly worldwide [1]. Between 2020 and 2050, population will double to reach 2.1 billion, presenting major challenges [1]. Of these 2.1 billion people, one subgroup will belong to a sexual or gender minority (or both), that is, individuals who identify as lesbian, gay, bisexual, transgender, or queer/questioning (LGBTQ) among others. The aim of this study was to review existing qualitative research involving older sexual minority adults in Japan and Sweden. International population-based studies investigating public health disparities have found a higher risk of both mental and physical health problems among sexual and gender minority adults [2,3] than among heterosexual cisgender people (cisgender people identify with the gender assigned to them at birth). This is explained by the higher stressor load generally experienced by sexual and gender minorities in the forms of stigma, discrimination, and minority stress [4,5]. Stressors are also experienced in the form of microaggressions, that is, more subtle social stressors in everyday life [6]. These are commonly experienced by youth and adult populations, including those in a Swedish context [7,8]. Lundberg et al. [7,8] highlight that besides being accustomed to hostile looks, misgendering and being asked personal questions, people who experience microaggressions in their everyday life perform ongoing emotional and relational work when navigating public and private spaces. Therefore, microaggressions drain energy and influence health and quality of life (QOL).

Older LGBTQ adults face unique challenges compared with their heterosexual cisgender counterparts [9,10,11]. In general, older LGBTQ adults commonly experience homophobia, transphobia, discrimination, and stigmatization throughout their lives [12]. Discrimination and victimization are associated with poorer physical and mental health in older LGBTQ adults [9]. Most studies on sexual and gender minority people do not include older adults, and most studies were only conducted with “older” adults in their 50s or even younger [9,13]. Additionally, most studies are conducted in Western countries, predominantly the English-speaking world, and studies from other parts of the world are lacking [2,14,15].

In the present study, we have focused on qualitative research to provide an in-depth understanding of the lived experiences and perceptions of older Japanese and Swedish LGBTQ adults. The reason for choosing Japan and Sweden is that the involved researchers have started a research project as part of a MIRAI collaboration, involving universities from both countries [16]. Japan and Sweden represent two distinct cultural contexts with different historical, social, and institutional backgrounds. Comparing these two countries allows for an exploration of how LGBTQ issues are understood, experienced, and addressed within contrasting cultural frameworks. This comparison can provide valuable insights into the cultural factors that shape the lives of LGBTQ individuals. In qualitative research, the focus is to understand and interpret social phenomena from the perspective of individuals or groups and in our review, this involves first-hand experiences from in-depth interviews [17]. Synthesizing findings from multiple qualitative studies can provide a wide range and depth of participants’ experiences and perspectives across contexts, as well as social and cultural aspects [18]. The rationale for choosing a qualitative approach, rather than a quantitative or mixed approach, in this case is as follows:In-depth understanding: By focusing on qualitative research, we can develop knowledge based on the details of participants’ experiences, capturing the richness and complexity of their stories.Contextual insights: By examining the social interactions and individual experiences in natural, non-experimental situations, we can gain valuable insights into the specific cultural, social, and historical factors that shape the experiences of older LGBTQ individuals in Japan and Sweden.Bridging research gaps: Synthesizing qualitative research findings can also help identify research gaps. By examining existing qualitative studies, we can identify areas where more in-depth investigations are needed and contribute to the development of future primary studies. This process can help fill the knowledge gaps and further advance our understanding of the experiences of older LGBTQ individuals.

Overall, the choice of a qualitative approach in this review is justified by its ability to provide a comprehensive, nuanced understanding of the experiences and perspectives of older LGBTQ individuals in Japan and Sweden, which is essential for informing policies, interventions, and healthcare practices [19].

Earlier qualitative systematic reviews of older LGBTQ adults have focused on specific topics, such as the experience of living with human immunodeficiency virus (HIV), ethnic minority elders, participants’ adaptive capacity, provider competency, determinants of health, stigma, isolation, needs, and support [9,20,21,22,23,24,25,26,27,28]. Common to these reviews is the difficulty in finding relevant studies for inclusion. This has led reviews to alter their inclusion criteria regarding age requirements; most commonly participants aged 50+ have been included [21,23], but some studies have included even younger participants [20]. In addition, some studies involved participants who did not identify as LGBTQ; it is unclear how this inclusion was justified in the analyses. Not all earlier reviews contained information about the settings of the included studies (country of origin); among those with this information, we found no Japanese and three Swedish studies [29,30,31]. Each of these three studies was included in separate reviews [11,25,26]. Overall, results show that older LGBTQ adults experience stigma, isolation, and stigma-related ill health, and that healthcare services are not equipped to cater to the specific needs of older LGBTQ adults [9,20,21,22,23,24,25,26,27,28]. Additionally, they also show that self-acceptance, social support, and respectful professionalism of caregivers can mitigate the negative impact of minority stress on health. As the inclusion criteria in the international reviews were often altered to include adults from 50 years or all adults, it is difficult to draw firm conclusions about the needs and experiences of older LGBTQ adults. Moreover, only few articles were found with strict inclusion criteria regarding age [20]. 

In this systematic review, we included LGBTQ adults aged 60 and above, following the definition of the World Health Organization (WHO) [32]. In the search, we focused on the following keywords: QOL, minority joy, and resilience and discrimination, stigmatization, and minority stress. The keywords are described briefly below.

Positive aspects were QOL, minority joy, and resilience. The WHO defines QOL as “an individual’s perception of their position in life in the context of the culture and value systems in which they live and in relation to their goals, expectations, standards, and concerns” [32]. In Japan, QOL has been described as living a life that is worth living and in accordance with subjective goals and expectations of life, even if it is restricted owing to health problems [33]. In Sweden, QOL refers to the ability of older adults to live with dignity, a sense of well-being, self-determination, participation, and security [34,35,36]. In Swedish law, the values set out for QOL articulate the ethical norms required for elderly care. The law declares that the goal is to ensure older adults’ need for dignity as well as to be able to live according to their identity and personality, to age with security and retained independence, live an active life in society, be treated with respect, and have access to good care and welfare [37]. Such care shall also be individualized, and services shall be of good quality. Minority joy refers to all positive experiences of people owing to their minority status. The term minority joy was coined by the author (MW) during a four-year study in Sweden, in 2021 [38]. Preliminary results from the study showed that minority joy includes factors, such as community connectedness, gender euphoria, and living an authentic life. Almost all participants described experiencing pride and strength because they had overcome challenges, which is consistent with existing literature on resilience. Resilience is the capacity to adapt in a positive way in the face of adversities [39]. That is, developing certain skills and strengths after experiencing stressors. Among older LGBTQ adults, resilience includes courage, self-reliance, and gender-role flexibility [40,41].

Negative aspects included in this review were discrimination, stigmatization, and minority stress. According to the Swedish Discrimination Act (2008:567) [42], discrimination occurs when a person is treated unfavorably or when their dignity is violated. The unfavorable treatment or the violation of the person’s dignity must be in connection to one of the seven grounds of discrimination, of which four were relevant in the context of this study: gender, gender identity or expression, sexual orientation, and age. The act of discrimination must also have taken place within an area of society where the law applies, for example, within healthcare or social services (Chapter 2, Article 13 of the Swedish Discrimination Act) [42]. Japan lacks a specific legal prohibition of discrimination on the basis of sexual or gender identity; however, courts have ruled that sexual orientation may be a factor under the existing discrimination law [43]. Importantly, what a person perceives as discriminatory may differ from the legal definition of discrimination, which, in turn, may differ between jurisdictions. In a study from the United States [12], LGBTQ adults 80 years and older reported experiencing discrimination and victimization. The most common lifetime discrimination described were verbal insults, followed by threats of physical violence. Stigmatization can be described as an individual’s experience of a negative attitude, or treatment from the greater society connected to a person’s identity or status, including the degree to which an individual self-identifies with a minority status group [44]. It can operate at both structural (i.e., institutions, healthcare) and individual levels, and can be seen as a public health issue because of its negative effects on the health and well-being of victims [45,46]. Closely related to stigmatization, and sometimes used interchangeably, is the term minority stress. It is defined as a chronic social stressor and includes the higher risk for stigma, discrimination, and stressors that minorities experience [47]. According to Meyer’s minority stress model [4], ill-health is not only a result of external factors, such as verbal insults and violence, but also of internal factors, such as expecting or fearing exposure to negative attitudes in society, internalized LGBTQ phobia, or hiding one’s identity. The model includes protective factors, such as coping and solidarity within the LGBTQ community. A population-based Swedish study of adults aged 18 years and older partially supported the minority stress model, because victimization and lack of social support were associated with higher risk for mental health problems [5].

An understanding of Japan and Sweden’s different culture and context is crucial when analyzing research of older LGBTQ adults in these countries [48]. Japan and Sweden have different legal frameworks and social attitudes towards LGBTQ rights and inclusion. Sweden is known for its progressive LGBTQ policies and has been at the forefront of promoting equal rights and protections. On the other hand, Japan has a more complex landscape with evolving attitudes and ongoing discussions surrounding LGBTQ rights. By comparing these two countries, it is possible to analyze the impact of legal and social factors on the experiences of LGBTQ individuals. Japan may be considered more socially conservative regarding gender roles than Sweden. However, the taboo against homosexuality was never as strong in Japan as in Sweden. Prior to Western influence in Japan during the Meiji period, it was, under certain circumstances, acceptable for men, particular among the aristocracy, to form sexual relationships with other men and gender-nonconforming people. Little is known of the attitudes toward female same-sex sexual activities and expressions. During the Meiji period, however, Western norms on homosexuality were imported, which led, among other things, to Japan’s only period of criminalization of same-sex sexual activities [43,49]. In the early 20th century, non-normative expressions of sexuality and gender were considered “perverse”. After the end of the second world war, LGBTQ communities began forming in Japan [49]. Sweden, in contrast, has historically and consistently regarded same-sex sexual activities and expressions as sinful [50], and, until 1944, illegal [51]. As such, gender and sexual minorities have been stigmatized during large parts of the 20th century, and continue to be, to a certain degree, in both countries. For example, a Japanese report showed that, in 2005, less than 14% of gay and bisexual men had come out to their parents, with experiences of familial homophobia common and attributed to Confucian-inspired heteronormative ideals [52]. Surveys on social acceptance of LGBT people show that on a 10-point scale, Swedish people score 7.9 (similar to Ireland and Nepal) and Japanese people 4.9 (similar to Hungary, Myanmar, and Bangladesh) [53].

Japanese law prohibits discrimination based on age, such as older adult abuse, including acts of physical abuse, neglect, psychological abuse, sexual abuse, and economical abuse [54]. In 2003, Japan established the Act on Special Cases in Handling Gender Status for Persons with Gender Identity Disorder (Act No. 111 of 2003), but there are strict requirements before being allowed to change legal gender; for example, individuals must not have a child under 18 years of age and they must visually pass as their desired legal gender. In addition to this possibility to change legal gender, the law and public system are not sufficient to protect the dignity of sexual and gender minorities. The question of how to end discrimination against sexual minority people has been discussed in the House of Representatives since 2016, but no law has been enacted yet [55]. For example, Japan does not permit same-sex marriage. Some local governments have issued ordinances recognizing same-sex partnerships, but they have no legal validity outside the remit of those ordinances.

In contrast, the Swedish law affords LGBTQ people relatively strong legal protections against discrimination based on sexual and gender identity in various sectors, such as in healthcare, the labor market, education, and housing. Protection against discrimination is stipulated in the Swedish constitution. The Swedish Instrument of Government underlines that no one may be disadvantaged because of their sexual orientation (Chapter 2, Article 12) [56]. The Swedish parliament extended hate-crime legislation to cover sexual identity in 2003, and gender identity and gender expression in 2019. In 1972, changing one’s legal gender became possible for trans people in Sweden if they identified as either a man or woman, following a process involving both medical and psychological examination (Legal Gender Recognition Act) [57]. The law was modernized in 2013 and a new gender identity law is expected in 2024, based on self-identification (the Government’s list of proposals) [58]. Management of healthcare also differs between the countries. In Japan, public healthcare is managed on a national and regional level, and care for older adults at home or in special accommodations is managed by the local governments. A significant aspect of Japanese healthcare is that specialized care for older adults is funded through taxation, social insurance fees, as well as by the users of the services (about 10%). The Japanese Ministry of Health, Labour and Welfare highlights that welfare services must preserve the dignity of the individual, including older adults´ sexual orientation and gender identity [59].

In Sweden, public healthcare is managed by regions, and care for older adults at home or in special accommodations is mainly managed by the municipalities. A significant aspect of Swedish healthcare is that specialized care for older adults is funded primarily through taxation, although tax-funded private alternatives in this sector are increasing rapidly. The Swedish government works, as part of the stated strategy, for equal rights and opportunities regardless of sexual orientation, gender identity, or gender expression, and healthcare and social services are identified as focus areas [51]. 

Male same-sex sexual acts were prohibited under Japanese law for 8 years, with criminalization through the Keikanritsujo-rei (the Sodomy Act) of 1872 and Kaiteiritsurei (the Supplemental Criminal Code) of 1873 until the implementation of the criminal code of 1880 which removed all provisions against same-sex sexual behavior, effectively legalizing it [43]. In 1944, Sweden decriminalized same-sex sexual activities, though Swedish authorities categorized homosexuality as a mental disorder until 1979 (Chapter 18 Section 10 the Swedish Penal Code [60]. Legal recognition for the LGBTQ group concerning family matters characterizes the last decades in Sweden; for example, the first act on same-sex cohabiting relationships was passed in 1988 [61], and a gender neutral, same-sex marriage code was passed in 2009.

Finally, to strengthen the situation for older adults in elderly care in general, a national value-based regulation was introduced in 2011 (Chapter 5, Article 4 of the Social Services Act [35]. It intended to create good conditions for older adults to live a dignified life with high well-being (QOL) within the publicly and privately operated elderly care in Sweden. A step in this progression is that some Swedish elderly care homes trained the staff on policy and routines so that they could obtain an LGBTQ-certification, and departments were opened exclusively for older LGBTQ adults.

In conclusion, older LGBTQ adults are facing several challenges and there is a lack of studies on this topic. It is unclear if the different levels of legislation in Japan and Sweden influence the experiences of older LGBTQ adults in the respective countries. Obtaining an overview of studies focusing on positive and negative experiences of older LGBTQ adults is crucial to be able to influence policy makers, to investigate potential factors influencing health, and to highlight the specific needs of older LGBTQ people.

The authors of this paper have formed a cross-national (Japanese–Swedish), cross-disciplinary research group to examine the situation of older LGBTQ adults. Through this study, we sought to illuminate the lived experiences of older LGBTQ adults in Japan and Sweden, and compare the differing legal, social, and cultural contexts in which older LGBTQ adults from the two countries may find themselves. In doing so, we also aimed to identify the gaps in research, which could help design further interventions and research projects in these countries to understand and advance the situations for older LGBTQ adults and better meet their needs in social and healthcare settings.

### 1.1. Aim

This systematic review aimed to analyze qualitative research on older LGBTQ adults conducted in Japan and Sweden with a focus on adults aged 60 and above. We attempted to synthesize the findings using a deductive approach focusing on QOL, minority joy, resilience, minority stress, discrimination, and stigmatization.

#### Research Question

What are the existing Japanese and Swedish qualitative studies on older LGBTQ adults, aged 60 or older, regarding QOL, minority joy, resilience, minority stress, discrimination, and stigmatization?

## 2. Materials and Methods

### 2.1. Study Design

A systematic literature review was conducted to summarize the existing knowledge regarding older LGBTQ adults´ experiences of QOL, minority joy, resilience, minority stress, discrimination, and stigmatization in Japan and Sweden. This review followed the Preferred Reporting Items for Systematic reviews and Meta-analyses (PRISMA) guidelines [62]. Predefined inclusion criteria were used when selecting articles according to the sample, phenomenon of interest, design, evaluation, research type (SPIDER) framework [63,64]. SPIDER was developed specifically for qualitative studies and is based on the same principles as the population, intervention, comparison, outcome (PICO) tool that is used for quantitative studies [65,66]. The Critical Appraisal Skills Program (CASP) checklist, developed by the Swedish Agency for Health Technology Assessment and Assessment of Social Services [67], was used to assess the quality of the articles, for more information see Appendix A. Finally, the results were summarized through a thematic synthesis approach [68].

### 2.2. Eligibility Criteria

The screening process included the eligibility criteria, shown in Table 1, and was predetermined in a study protocol (see Appendix A). All qualitative studies with first-hand material from adults aged 60 years and older were included if they met the inclusion criteria.

### 2.3. Search Strategy and Study Selection

Author AB conducted preliminary searches with the guidance of a specialist librarian at Linnaeus University and collected keywords from relevant articles written by experts in the field of older LGBTQ adults, in combination with database-specific search terms (thesaurus). Only peer-reviewed articles were searched from the selected databases. Search terms were chosen based on the preliminary search by AB and the librarian. For more information about the search strings, refer to the review protocol in the Appendix A.

Author TM conducted preliminary searches in the database of the Kyushu University Library and collected keywords from relevant articles written by experts in the field of older LGBTQ adults, in combination with database-specific search terms (thesaurus).

The first Japanese search was conducted in September 2021 and the first Swedish searches were conducted in January 2022. The last searches, both Japanese and Swedish, were conducted in March 2023.

We also searched for grey literature on Google Scholar using the term “LGBTQ older adults” followed by searches combining this term with separate key-words (QOL, resilience, minority stress, discrimination, and stigmatization). The first 200 results per search were reviewed (a review of 1000 search results in total). This search yielded no new articles for inclusion.

TM screened all the Japanese results. AB screened all the records retrieved from the international searches, which included articles from Japan and Sweden and the Swedish searches. 

### 2.4. Synthesis Methodology

We summarized the results using a thematic synthesis [68]. The analysis was conducted in three stages: (1) coding the text line-by-line, (2) developing descriptive themes; and (3) creating analytical themes. In Step 1, we read and re-read the findings of all the four included articles. AB and RH coded each line of text independently, and codes were first created inductively. When reading the text line-by-line, codes were created based on the meaning and content of the data. New codes were developed, when necessary. In Step 2, we created descriptive themes that captured the meanings of a group of initial codes. A draft summary was written by AB and RH as a synthesis similar to the original findings. The other authors commented on this draft. In Step 3, we used a deductive approach to complement the inductive findings and synthesized the data by creating themes based on the keywords (QOL, minority joy, resilience as well as discrimination, stigmatization, and minority stress). In this step, we generated a synthesis that addressed the aims of the study and, thus, created analytical themes, going beyond the original studies. Four of the authors are experienced in conducting qualitative syntheses.

## 3. Results

A total of 1191 studies were retrieved, of which 431 were duplicates. Thus, 760 papers were screened (Figure 1).

### 3.1. Data Extraction

One Japanese peer-reviewed paper was found [70]. The age range of the participants was 20 to 70 years, which did not match the protocol of this study. Nevertheless, the themes found in the paper were similar to those found in the Swedish papers: “end-of-life period and post-mortem arrangements,” “support services and consultation agencies,” “participation in the community and support systems,” “connecting with people,” “anxieties about long-term care,” “thinking about one’s life-plan,” and “a vague sense of anxiety.” There were no peer-reviewed Japanese articles to be included in the synthesis. 

Five Swedish articles were found in the systematic literature search (see Table 2). Four articles were included in the synthesis. We excluded one article [29] because the analysis was not distinctive for older LGBTQ adults but focused on aging-related topics for older adults in general. Therefore, four articles were included in the synthesis. For more information about the quality assessment, see Appendix A.

### 3.2. Findings

In total, data from 48 participants aged 60–94 years were included. Three articles had overlapping samples, as specified in Table 2. The results are presented in two themes: (1) *QOL, minority joy, and resilience* (positive aspects), and (2) *experiences of discrimination, stigmatization, and minority stress* (negative aspects). 

#### 3.2.1. QOL, Minority Joy, and Resilience

QOL results from being able to come out and be recognized and accepted for one’s identity [29,73]. Some of the transgender participants in Siverskog’s study [29] experienced more freedom in exploring their identities after retirement. The participants described a desire to be acknowledged and recognized for their own qualities and unique life history, while also being treated like everyone else, as one lesbian participant in Löf and Olaison’s study [73] said:


*… We’re all individuals and we’re all different. You have to show consideration to everyone.*



*And some people need to act or think in one way. And other people think in another way. And that means you can’t treat everyone the same because it’s wrong somehow…*


Before disclosing one’s gender or sexual identity, one needs to feel safe and respected as an older LGBTQ adult. The participants in Löf and Olaison’s study [73] emphasized that to feel safe in a future care situation, nursing staff need to have LGBTQ knowledge and be able to treat older LGBTQ adults in a competent and affirmative way.

Experiences of minority joy were found in the narratives about “coming in” to an LGBTQ community [72]. Here, “coming in” refers to joining a community, and “coming home” refers to feeling at home within the community. Memories of “coming in” also relate to the mysteriousness of the LGBTQ community wherein different rules apply and some places are inaccessible and hidden from the rest of society [72]. “Coming in” is used in contrast to “coming out [of the closet],” a metaphor used within the LGBTQ community to describe self-disclosure of sexual orientation or gender identity to other people. “Coming in” and finding collective strength together with people with shared experiences was described as life-changing and as providing a sense of liberation. As one participant in Siverskog and Bromseth´s study [72] said:


*It has strengthened me a lot. When I meet my old classmates, even if they understand that I am gay, I feel that I have my own life, I belong somewhere, and I have my own context and that is so very important; so I am really happy for the struggle that has been faced*


Some participants in Löf and Olaison’s study [73] discussed how important it was to have a sense of belonging with others who have had the same experiences throughout life and had lived a life not in accordance with societal norms, and that they would prefer to live in a special housing for older LGBTQ adults. 

The LGBTQ community has shown resilience in response to memories of struggles, where people have come together and joined each other in fighting for a common cause (LGBTQ rights) [72]. One example of this is the HIV/AIDS crisis of the 1980s and 1990s, wherein the community collectively cared for one another and experienced solidarity, but many also had personal memories of loss and grief [72].


*We had several people [in our gay choir] who were HIV positive and died. And some of them got so skinny. We sang at the Rosenlund hospital for those who were hospitalized there, and for some, it was so secretive that they did not even come out of their rooms but just opened the door so they could hear us. Their parents had no idea they were HIV positive. It was pretty horrible.*


However, for some participants, the sense of community had eroded, and they felt alienated because of their age when accessing LGBTQ community events that largely catered to and centered around younger members. Some of the participants in Siverskog and Bromseth´s study [72] had no access to LGBTQ spaces and had little social support.

#### 3.2.2. Discrimination, Stigmatization, and Minority Stress

One participant, who had identified as a woman during a large period of his life, had worked as a teacher and participated in a documentary on national TV. Afterwards, the participant was harassed by colleagues and was offered to retire earlier than planned [29]. Male, today stealth with trans praxis, described:


*They [colleagues] used me as a subject for the morning prayers, praying for me to get cured… I experienced more and more opposition at work and then I got called up to my boss, who offered to retire me early.*


Several transgender participants in Siverskog’s study [29] had experienced stigmatization from an early age and throughout their lives. When exhibiting gender-non-conforming desires, such as dressing according to the other legal gender’s norms, they felt that such behaviors were not socially acceptable. Some described being shamed or even abused when their trans identity was discovered. With time and growing up, shame related to dressing or acting according to the other legal gender increased, and resulted in hiding, being careful, and so on [29]. As one participant who identifies as a man but has a female gender expression full time said:


*It has been a threat that someone would find out that I was interested in wearing women’s clothes. So I stayed away from that. I felt I would be completely estranged and left out if I did that. And that is something transvestites live with to a great extent; that you simply get pointed out and shamed. And I lived with this, and still do, as a limiting part.*


Although the concept of minority stress was not explicitly formulated in the studies, there were several descriptions of situations where the participants experienced stressors associated with their sexual minority status. For example, the older trans people experienced a lack of knowledge regarding trans in different contexts [29]. When meeting cis members of the LGBTQ community, the trans participants found that there was a lack of basic trans understanding [29]. Additionally, the participants experienced a lack of knowledge in the healthcare sector, even in healthcare professionals who were specialized in trans care (gender affirmative care). They had to educate their doctors, care staff, and social workers. One participant brought information material to her doctor. Some participants described having to come out repeatedly in new healthcare relationships (physicians, and hormone replacement therapy nurses) and having to explain trans-related issues [29]. Some of them feared being labeled or seen as disgusting by healthcare providers [29]. One transsexual woman described her thoughts:


*As long as I live at home, I think it will be alright, but then when you get older and maybe have to move to a nursing home … Yes, when it is time and they come here and see that it is a man in women’s clothes, “God how disgusting; we don’t want to go to that person again,” you know right?*


Some of the participants in Siverskog’s study [71] found aging problematic in relation to performing gender, while others did not. LGBTQ spaces were harder to access because of ageism and the participants lacked social support, which led to feelings of loneliness and isolation. With age, loss as well as lack of bodily functions are normalized, as such, cis bodies and trans bodies converge in function and appearance, as compared to younger bodies [29]. Simultaneously, older age can make gender-confirming surgery riskier and more difficult—one’s own body making such surgery impossible. If the culture had been more accepting, some participants would have sought out surgery earlier in their younger years. Some regretted not having come out earlier [71]. For those who had age-related health reasons to not undergo gender reaffirming surgery, coming out late also influenced a legal gender change, because the law, at the time of data collection, mandated sterilization for a change of legal gender [71]. One transsexual woman said:


*… when they heard that I had atrial fibrillation and a pacemaker and took a lot of heart medicine and Varan [medicine], yes, then [the doctor] just put down the pen and said, “you can forget about that, because no one will put a knife in you if it’s not absolutely necessary”*


The quote from the doctor suggests a view that gender reaffirming surgery for her was “not absolutely necessary” (regardless of whether medically possible or not), leading to stigmatization of the participant´s transsexual identity.

## 4. Discussion

With this systematic review, we sought to evaluate and synthesize qualitative research on the lived experiences of older LGBTQ adults (aged 60 and above) in Japan and Sweden. We synthesized the findings focusing on QOL, minority joy, resilience, minority stress, discrimination, and stigmatization. With no study from Japan and only four Swedish papers to review, the findings were summarized in two over-arching themes: (a) QOL, minority joy, and resilience, and (b) discrimination, stigmatization, and minority stress.

The results of this systematic review were limited. While we found one peer-reviewed article from Japan, it did not solely include older adults, and therefore we excluded it. Five peer-reviewed journal articles from Sweden (2014–2020) focusing on LGBTQ adults aged 60 years or older were found, of which one was excluded since the phenomenon was not distinctive for older LGBTQ adults. Of these remaining four, three articles used overlapping data sets and were flagged as having quality concerns. These issues highlight the fact that older LGBTQ adults are an understudied population and there is a need for more studies in the field. Even though the number of papers about LGBTQ older adults has gradually been increasing from around 2015 in Japan, most of them are not peer reviewed, or do not include the right age span, or first-hand experiences from the LGBTQ older adults themselves. Furthermore, it seems that the subject has not been established as a research field in Japanese academia yet. Researchers may not have recognized this because of the lack of information about them, and there are also difficulties finding older LGBTQ participants for the research. Looking at the current Japanese society, the awareness about LGBTQ issues is increasing, but people’s attention is primarily focused on the situation for the younger generation. There are some statements in non-peer reviewed papers about older LGBTQ adults’ difficulty to come out about their sexuality in Japanese society because of the strong stigma [74,75,76,77]. In addition, older LGBTQ people report that they, since youth, have been treated as invisible people in Japanese society [76,78,79].

The selected articles did not specifically study QOL, resilience, or minority joy, but we found aspects in the studies relevant to living a life with dignity ensuring wellbeing (QOL), overcoming difficulties in a positive way (resilience), and experiencing positive aspects due to identifying as LGBTQ (minority joy). Participants described that it was important to be recognized and accepted for one’s identity, be able to talk about oneself, receive respect from care staff, and obtain care for individual needs [73]. For some participants, older age seemed to be associated with more freedom in exploring one’s identity [29,71,72]. Resilience and minority joy were primarily described in relation to community and to finding a home in LGBTQ spaces. Participants highlighted the importance of finding people with similar experiences. Difficult experiences, such as the HIV pandemic, which was related to both personal and communal grief, also gave rise to activism, solidarity, and communal caretaking, which were described in positive terms. The exploration of positive aspects related to older LGBTQ adults have been studied to a lesser degree than minority stress and negative outcomes. The results of this study are largely consistent with preliminary results from the Swedish study on minority joy [36], and earlier reviews concerning older LGBTQ people. For example, Averett and Jenkins [21] and McCann and Brown [27] reported participants’ resilience in handling challenges, and McParland and Camic [26] highlighted community as an important health factor for older LGBTQ adults.

One serious experience of discrimination in the legal sense was described, wherein the participant faced harassment form colleagues, and several (especially trans) participants had experienced stigmatization from an early age and throughout their lives [22]. Minority stress was described as associated with low knowledge about LGBTQ issues in society at large and in healthcare and care facilities [22]. For transgender participants, this was also experienced in LGBTQ spaces. This is in line with earlier studies on youth and adults in a Swedish context [7,8]. Some participants described how LGBTQ spaces were harder to access because of ageism and that they lacked social support, which led to feelings of loneliness and isolation. Moreover, some participants also experienced age-related difficulties when trying to access gender affirming care [73]. Loneliness is described in earlier reviews on older LGBTQ adults [20,24], and negative attitudes and discrimination, including negative treatment in residential or other care settings, are well studied [20,22,26]. The results from the current study are, therefore, consistent with those of earlier studies in the area, mainly conducted in English-speaking countries.

In contrast to earlier reviews, which included studies focused primarily on gay men, followed by lesbians and bisexuals, and only few on transgender participants [9,21,22,23,24,26], the Swedish studies tended to focus more on interviews with trans people. Therefore, trans-related topics, such as gender-affirming healthcare and trans-specific stigma, were prevalent in the synthesis of the included studies.

Japan and Sweden share similar demographic structures concerning a growing population aged 60 years or over [80]. Nevertheless, they differ immensely and have different socio-legal starting points, with Sweden having greater social acceptance toward LGBTQ people than Japan [53]. The articles used in the synthesis reveal difficulties for LGBTQ adults in Sweden to assert the same rights as heterosexual and cisgendered people and difficulties living in society with a high QOL. All articles reviewed specified legal conditions regarding discrimination and QOL with direct references to current and relevant legal sources. Thus, this synthesis reveals the current scenario of the problems that older LGBTQ adults experience in contact with society. However, it is noteworthy that although the articles reported experiences of perceived discrimination among the participants in the study, they did not analyze or discuss if such experiences meet the legal criteria of discrimination in the legal sense [81].

### 4.1. Study Limitations

This study has some limitations. There is limited qualitative research on older LGBTQ adults in Japan and Sweden, and the current study was based on four Swedish articles only. Therefore, it is not possible to draw any generalizable conclusions about the needs of older LGBTQ people based on these limited findings. Second, although we assessed the quality of the articles, we included articles despite concerns regarding the methods used. This was done because of the difficulties in finding relevant literature. Thus, qualitative researchers must ensure a high degree of quality in their work; for example, declaring a clear and replicable methodology and ensuring validation of interpretations. At the same time, we must acknowledge the differing methodological cultures across disciplines. Therefore, the strict inclusion criteria of this study, although intended as a strength, may be seen as a weakness. Had we adhered fully to the inclusion criteria, we would have been able to include only one study in the synthesis. Finally, although we sought to illuminate the lived experiences of older LGBTQ adults in Japan and Sweden, and thereby compare the differing legal, social, and cultural contexts, the present results can relate only to the Swedish circumstances. Thus, the Japanese circumstances cannot be inferred owing to the lack of research on older LGBTQ adults. However, TM and YH, the Japanese authors, conducted a comprehensive review of the Japanese research literature, which included various types of papers, such as opinion papers. This review identified several important themes related to addressing LGBTQ issues among older individuals. The findings indicated the need for further qualitative studies that involve the opinions of affected individuals and relevant agencies to address these issues. Eleven informative papers were found during the Japanese search, out of which one was peer-reviewed and the remaining ten were non-peer-reviewed. These papers were excluded from the current study because they did not align with our research protocol, for more information see Appendix A, which focused on individuals aged 60 and above and specifically included peer-reviewed publications as well as first-hand experiences of LGBTQ older adults. Considering the Japanese academic landscape in this research area, we nevertheless decided to explore non-peer-reviewed papers to gather significant information about LGBTQ older adults and develop hypotheses for future studies in this field. The results from these non-peer-reviewed papers will be published in a separate article in the near future.

Additionally, during the course of this study, the Japanese authors consulted healthcare professionals regarding LGBTQ issues in healthcare settings. Surprisingly, none of the professionals recognized the significance of such issues or acknowledged their existence based on their clinical experiences. Consequently, it is crucial for Japanese researchers to conduct further research that directly involves the perspectives of older LGBTQ individuals who have been affected by these issues.

The situation for older LGBTQ adults in Japan is severely understudied, which is in line with the situation in Asia in general. For example, a previous mixed methods scoping research on older LGBTQ adults in Asia found only 10 articles for inclusion (4 were qualitative; of all the articles, 5 were from Israel, 2 from the Philippines, and 1 each from China, India, and Thailand) [15]. More research is required in Japan and across Asia.

### 4.2. Strengths

This qualitative synthesis demonstrates several strengths. First, as we had strict inclusion criteria, we ensured that the views of only older LGBTQ adults (and not others) were part of the synthesis. Second, the reviewers were blind to one another at each stage of the review, which ensured that bias or undue influence from other team members could be minimized. This suggests that the findings are, as far as possible, impartial and objective. Finally, we discussed diverging interpretations until consensus was reached, which indicates that we undertook a collaborative and thorough approach to the analysis. Therefore, despite the demonstrated weaknesses, the synthesis should be regarded as trustworthy.

This synthesis highlights the need for more rigorous qualitative research, especially in Japan, but also in Sweden, on the context and experiences of older LGBTQ adults.

## 5. Conclusions

To the best of our knowledge, this synthesis is the first to include only Japanese and Swedish articles. While we could not include any Japanese articles, the synthesis highlights the Swedish context and the need for more research on older LGBTQ adults.

The Swedish articles showed that participants wished to be acknowledged for their identities, unique life histories, and assets. They wished to be treated like everyone else, while emphasizing the need for staff in healthcare services to possess a greater understanding and respect for the unique circumstances of older LGBTQ adults.

## Figures and Tables

**Figure 1 ijerph-20-06281-f001:**
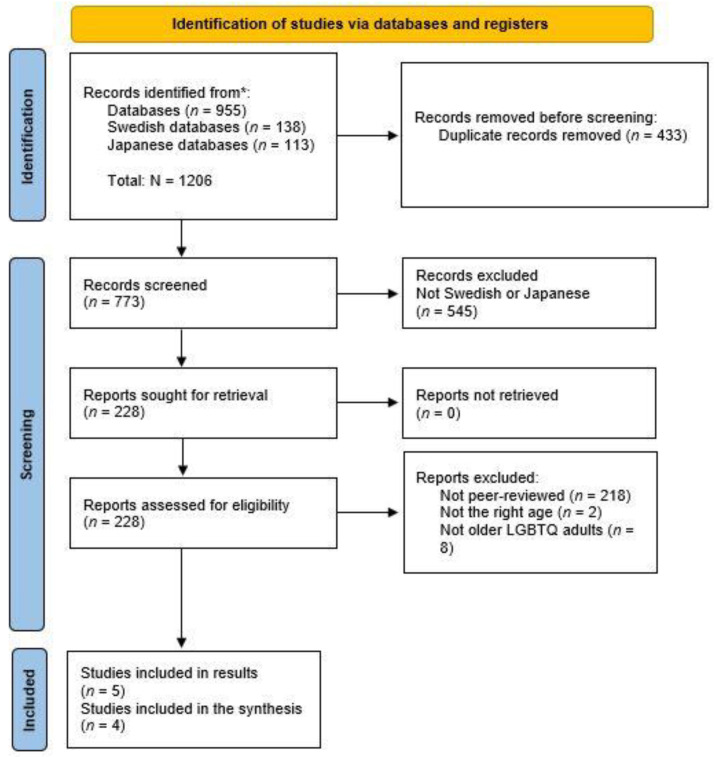
Flow chart. * Searches for both Japanese and Swedish articles: ASSIA (*n* = 48); CINAHL (*n* = 189); PsychINFO (*n* = 420); Medline (*n* = 248); Social Abstracts (*n* = 50). Swedish articles only LIBRIS (*n* = 41) SwePub (*n* = 26) DIVA (*n* = 71); Japanese articles only: CiNii Articles (*n* = 27); Japan Medical Articles Society (*n* = 71); PubMed (*n* = 11); Scopus (*n* = 4). Adapted with permission from [69].

**Table 1 ijerph-20-06281-t001:** Inclusion criteria organized according to SPIDER framework.

Sample	Phenomenon of Interest	Design	Evaluation	Research Type
Japanese or Swedish	Quality of life, minority joy, resilience, discrimination, stigmatization, and minority stress	Focus groups, and interviews	Experiences, feelings, attitudes, perceptions, and views	Qualitative research
LGBTQ adults aged 60 years and above				Written in Japanese, Swedish, or English

Notes: SPIDER: sample, phenomenon of interest, design, evaluation, research type; LGBTQ: lesbian, gay, bisexual, transgender, or queer/questioning.

**Table 2 ijerph-20-06281-t002:** Articles extracted in the review.

Article no.	Title	Country	Context/Data Collection	Sample/Age	Phenomenon of Interest/Research Question(s)	Methodology/Analysis	Results	Quality	Included or Excluded	Specified Legal Prerequisites (Quality of Life and Discrimination)
1. [31]	Turning vinegar into wine: Humorous self-presentations among older LGBTQ online daters	Sweden	Two web-based forums in Sweden were used for data collection. The first forum was directed at “homosexual, bisexual, queer and trans people along with their friends,” and the second at “homosexual and bisexual girls and women.” Data were collected using the automatic search functions of the forums.	*n*= 276; male, *n*= 162; female, *n*= 88; transgender, *n* = 26. Age: 60–81 years.	Whether self-mocking comments about old age and age-related topics confirm or subvert prevalent norms and images related to age and sexuality.	Quantitative content analysis. Data were sorted according to the profile contents, stated as attributes concerning personality, interests, body/appearance, education/career, comments on age, sexual content, nude photographs, and mentions of ethnicity as well as humor. The empirical analysis was divided into two sections: (1) self-mocking comments as a form of “age-salient maneuvering” related to existing age norms, and (2) self-mocking comments about old age, gray hair, wrinkles, being overweight, and impotence as a way of performing marketable characteristics, such as humor, self-distance, and honesty.	Themes: 1. Humorous comments on age-related issues. 2. Is humor subversive or conservative? 3. Self-mocking comments used as age-salient maneuvering.	High concerns	Not included. The analysis was not distinctive for the LGBTQ group, and the comments were not related to age concerns in general.	Insignificant concerns; legal prerequisites of minor importance.
2. [29]	“They Just Don’t Have a Clue”: Transgender Aging and Implications for Social Work	Sweden	Recruitment via newspaper ads, snowball sampling, and an online LGBT community. The sample was collected from a larger project including 20 interviews with older LGBTQ people.	Transgender, *n* = 6 Age: 62–78 years.	How earlier life experiences matter in later life, and how age and (non-conforming) gender identities are understood in relation to one another.	Thematic analysis (Braun and Clarke, 2006). The participants were encouraged to talk freely about their lives, starting with when and where they were born. They were asked to follow-up from their stories, concerning their gender identities, social networks, relations, health, aging, and the body during different periods of their lives.	Themes: 1. Intersections of age and gender during the course of life. 2. The lack of knowledge on transgender issues within different contexts. 3. How previous experiences of accessing care and social services matter in later life and in relation to the future need for care.	High concerns. The analysis is unclear, only one researcher conducted the analysis, reflexivity and how the findings were validated are not described.	Included. The findings are relevant and there is a lack of studies in the field.	Insignificant concerns; relatively strong legal framework (discrimination).
3. [71]	Ageing Bodies that Matter: Age, Gender and Embodiment in Older Transgender People’s Life Stories	Sweden	The same sample as in [29] the project in which six trans-identified persons were included in a sample of 20 older LGBTQ identified adults.	Transgender, *n* = 6 Age: 62–78 years.	How gender, age, and embodiment intersect in relation to trans identity, and what old age and aging mean for transgender people.	Thematic analysis (Braun and Clarke, 2006).	Themes: 1. Material bodies focus on the physical body: how it matters in the performance of (linear) gender and how it can fail in relation to the desire to “pass,” and how age and aging play into this experience. 2. Performing gender and age. Bodily aging can be perceived differently depending on bodily conditions and one’s ability and need to perform gender.	High concerns	Included	Insignificant concerns; relatively strong legal framework (discrimination).
4. [72]	Subcultural Spaces: LGBTQ Aging in a Swedish Context	Sweden	The article includes two sub-studies: (1) an ethnographic study based on participant observation and 13 interviews; all participants lived in the Stockholm area and were involved in subcultural communities (lesbian feminist or LGBTQ) in different degrees and with various engagements. (2) the study described in [29,71], is based on interviews with 20 people identifying as LGBTQ.	Sample 1: Non-heterosexual cis and trans women, *n* = 13. Age: 60–94. Sample 2: LGBTQ, *n* = 20; trans, *n* = 6. Age: 64–88 years	Experiences of community among older LGBTQ people. The processes of finding, entering, and creating subcultural spaces. The influence of time and geographical context on these experiences. Aging within these communities?	Thematic analysis (Braun and Clarke, 2006).	1. Coming in, coming home: finding spaces of belonging. 2. Spaces with friction: uncomfortable spaces. 3. Aging, bodies, and community: continuity and change.	High concerns. Recruitment and data collection were not clearly specified, concerns regarding analysis of the material, and standards of thematic analysis were not followed.	Included	Insignificant concerns; relatively strong legal framework (increased civil rights and discrimination.
5. [73]	“I do not want to go back into the closet just because I need care”: recognition of older LGBTQ adults in relation to future care needs	Sweden	Participants were recruited via pensioners’ organizations; LGBTQ organizations, including those for older LGBTQ adults; one LGBT senior housing facility; and LGBT-certified retirement homes/home care services. Some were recruited through a Pride festival. In addition, a snowball sampling procedure was used.	*n*= 15, bisexual/lesbian women, *n* = 5; bisexual/gay men, *n* = 5; transgender, *n* = 5 Age: 65 years and older The interviewees lived in large and medium-sized cities in north and south Sweden. The interviewees lived at home and two of them had previous experience with elder care services. Three lived in an LGBT senior housing facility.	How older Swedish LGBTQ adults reason about openness in an elder care context concerning their future needs for services.	Thematic approach (Braun and Clarke, 2006).	Themes: 1. Openness and recognition 2. Preferences regarding how to be treated in elder care 3. LGBTQ housing	Insignificant concerns. Recruitment and analysis were clearly described; however, reflexivity was not discussed.	Included	Insignificant concerns. Concerns about the lack of considerations regarding the legal framework (the right of equal treatment, QOL, etc.)

## Data Availability

Not applicable.

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
