# Peer review of "A Systematic Review of Qualitative Research Literature and a Thematic Synthesis of Older LGBTQ People’s Experiences of Quality of Life, Minority Joy, Resilience, Minority Stress, Discrimination, and Stigmatization in Japan and Sweden"

_ijerph, 2023, doi:10.3390/ijerph20136281_

Round 1

Reviewer 1 Report

Thank you for sharing this article. I have a few minor comments.

1. I am wondering why this study only focused on qualitative findings, especially when there are no Japanese studies available in the literature. 

2. Would you comment on why there are no Japanese studies? Is this possibly because this population has been stigmatized and underrepresented?

3. Is it useful to expand the regional focus to East Asia instead of Japan?

4. What would be the direction of future research? How should Japanese LGBTQ researchers move forward based on the findings of this research? What kind of funding is available? What kind of partnership is needed? 

Author Response

We thank you and the reviewers for your thoughtful suggestions and insights. The manuscript has benefited from these insightful suggestions. I look forward to working with you and the reviewers to move this manuscript closer to publication in the International Journal of Environmental Research and Public Health.

The manuscript has been rechecked and the necessary changes have been made in accordance with the reviewers’ suggestions. The responses to all comments have been prepared and attached herewith.

We reply to each comment and mark changes in the manuscript with yellow:

Reviewer 1

  1. Reviewer comment: I am wondering why this study only focused on qualitative findings, especially when there are no Japanese studies available in the literature.

Reply: Before the study started, we made a protocol of the study, which we have followed. We wanted to address older LGBTQ people’s in-depth life experiences, and we did not want to mix quantitative and qualitative studies, and also not include other age groups than participants over the age of 60. The reason for using qualitative studies only is clarified, page 2, line 66. The results show that there is a need for more qualitative research in both Japan and Sweden, and that Japan have qualitative research but not-peer reviewed. These qualitative studies will be published in another article, because we do not want to mix peer-reviewed and non-peer-reviewed studies in this systematic review. This is as wan important criteria for inclusion in our study protocol.

  1. Reviewer comment: Would you comment on why there are no Japanese studies? Is this possibly because this population has been stigmatized and underrepresented?

Reply: The main reasons for that there were no Japanese studies included was that the studies found were not peer reviewed and that there were not first-hand responses from older LGBTQ people themselves. We discuss this matter more, page 11, line 483.

  1. Reviewer comment: Is it useful to expand the regional focus to East Asia instead of Japan?

Reply: This was not the purpose of this study. Legal conditions differ between countries, so it is crucial that legal rules and the structure of legal institutions are explained in the article. Comparisons with other countries (in East Asia [China] or in Europe [Russia], cannot therefore be made without a study of the legal preconditions. In this study, fundamental legal concerns are clarified about the positive and negative aspects the LGBTQI group faces in Japan and Sweden. [Lasser, Mitchel de S.-O.-l’E., (2003) The question of understanding, In Comparative Legal Studies: Traditions and Transitions, Pierre Legrand and Rodrick Munday, (eds.) Cambridge University Press, Cambridge, p. 197-239 and Legrand, Pierre, (2003) The same and the different, In Comparative Legal Studies: Traditions and Transitions, Pierre Legrand and Rodrick Munday, (eds.) Cambridge University Press, Cambridge, p. 240–311.] 

  1. Reviewer comment: What would be the direction of future research? How should Japanese LGBTQ researchers move forward based on the findings of this research? What kind of funding is available? What kind of partnership is needed?

Reply: This issue is discussed in more detail in the discussion section, page 13, line 570.

Reviewer 2 Report

This study uses systematic review of existing literature and qualitative research among 48 participants in Sweden and Japan, to identify possible positive and negative sides faced by LGBTQ. It is found discrimination, minor stress, lack of social and legal support, especially in aged care are their major barriers in their life.  

As discussed by the authors, the limitation of this study is that the literature are mainly in Swedish circumstance, representing a western cultural context. More study will need to be conducted in Asian cultural context, such as the support from their parents and siblings.

I would suggest authors distinguish the findings by western and Asian cultural contest, what are their common barriers and what are different in different culture.

In addition, the formating of Tables 1 and 2 need to be improved. . 

Author Response

We thank you and the reviewers for your thoughtful suggestions and insights. The manuscript has benefited from these insightful suggestions. I look forward to working with you and the reviewers to move this manuscript closer to publication in the International Journal of Environmental Research and Public Health.

The manuscript has been rechecked and the necessary changes have been made in accordance with the reviewers’ suggestions. The responses to all comments have been prepared and attached herewith.

We reply to each comment and mark changes in the manuscript with yellow:

Reviewer 2

Reviewer: This study uses systematic review of existing literature and qualitative research among 48 participants in Sweden and Japan, to identify possible positive and negative sides faced by LGBTQ. It is found discrimination, minor stress, lack of social and legal support, especially in aged care are their major barriers in their life.  

As discussed by the authors, the limitation of this study is that the literature are mainly in Swedish circumstance, representing a western cultural context. More study will need to be conducted in Asian cultural context, such as the support from their parents and siblings.

  1. Reviewer comment: I would suggest authors distinguish the findings by western and Asian cultural contest, what are their common barriers and what are different in different culture.

Reply: We discuss more about the cultural aspects in the introduction as well as the discussion section. Page 4, line 176 and page 12, line 553.

  1. Reviewer comment: In addition, the formatting of Tables 1 and 2 need to be improved.

Reply: the formatting of the tables have been improved.

Reviewer 3 Report

An interesting paper.  But there are some serious limitations.

First, the international literature needs to be better represented and you need to show what is already known, and what new contributions are shown by your paper.  Below are some research work reading as part of your review.

Second, you need to better describe how you define qualitative research.

Third, a better justification or rationale for selecting these two countries is required.

Fourth, can you convince the reader that your design has validity and you have captured all the Japanese literature published in English on this topic?

Finally, the paper lacks a proper sample size of  studies to make a significant contribution to the literature.

some literature worth reading 

Reflecting on Life Then and Now: Interviews on the Life Courses of Older Lesbian Women and Gay Men in Australia

Springer

  • October 2021
  • Sexuality Research and Social Policy 20(6):1-14

DOI:10.1007/s13178-021-00653-z

Volunteering among Older Lesbian and Gay Adults: Associations with Mental, Physical and Social Well-Being

  • August 2020
  • Journal of Aging and Health 33(1-2):089826432095291

Older lesbian and gay men’s perceptions on lesbian and gay youth in Australia

  • February 2020
  • Culture Health & Sexuality 23(1):1-16
DOI:10.1080/13691058.2019.1696984

Recruiting stigmatised populations and managing negative commentary via social media: a case study of recruiting older LGBTI research participants in Australia

  • December 2020
  • International Journal of Social Research Methodology 25(10)

DOI:10.1080/13645579.2020.1863545

Author Response

We thank you and the reviewers for your thoughtful suggestions and insights. The manuscript has benefited from these insightful suggestions. I look forward to working with you and the reviewers to move this manuscript closer to publication in the International Journal of Environmental Research and Public Health.

The manuscript has been rechecked and the necessary changes have been made in accordance with the reviewers’ suggestions. The responses to all comments have been prepared and attached herewith.

We reply to each comment and mark changes in the manuscript with yellow:

Reviewer 3

  1. Reviewer comment: First, the international literature needs to be better represented and you need to show what is already known, and what new contributions are shown by your paper. Below are some research work reading as part of your review.

Reply: Earlier reviews in the field are described at page 3, line 98.

  1. Reviewer comment: Second, you need to better describe how you define qualitative research.

Reply: How we define qualitative studies is clarified, page 2, line 66.

  1. Reviewer comment: Third, a better justification or rationale for selecting these two countries is required.

Reply: The reason for choosing Japan and Sweden is that the involved researchers have started a research project as part of a MIRAI collaboration, involving Universities from both countries. However, we believe that an in-depth comparison between Japan and Sweden is of interest for a wider research community. We describe this at page 2 and 4, marked in yellow.

  1. Reviewer comment: Fourth, can you convince the reader that your design has validity and you have captured all the Japanese literature published in English on this topic?

Reply: Yes, we think that our searches for over a year in the topic is according to all guidelines in how to conduct a systematic review. Information can be found in our protocol, see supplementary file we have described our searches in both English, Swedish and Japanese literature, see page 7, line 310. The issue is discussed more thoroughly at page 11, line 497.

  1. Reviewer comment: Finally, the paper lacks a proper sample size of studies to make a significant contribution to the literature.

Reply: We emphasize that it is not possible to draw any conclusions based on the limited number of participants page 13, line 589. However, we believe our findings is a result per se, that the number of studies is so limited. And since, as far as we know, there are no other studies comparing Japan and Sweden in this context it is of interest to a wider research community in the field.

Thanks for recommending the interesting articles!